# Effects and Mechanism Analysis of Non-Bagging and Bagging Cultivation on the Growth and Content Change of Specific Substances of Fuji Apple Fruit

**DOI:** 10.3390/plants12183309

**Published:** 2023-09-19

**Authors:** Guiping Wang, Ru Chen, Xueping Han, Xiaomin Xue

**Affiliations:** Shandong Institute of Pomology, Taian 271000, China; guigui-0530@163.com (G.W.); chenru.8668@163.com (R.C.); hanxuepingrun@163.com (X.H.)

**Keywords:** Fuji, non-bagging, sugar acid metabolism, mineral elements, nutrients, antioxidant capacity, gene expression

## Abstract

The experiment was conducted at the Taidong Base of Shandong Institute of Pomology, Tai’an City, Shandong Province, China, from May to October 2022. Using Fuji Apple Tianhong.2/SH/*Malus robusta* (*M. robusta*) as experimental materials, the differences and mechanisms of the effects of non-bagging and bagging cultivation on the growth and changes in some substance content of Fuji apple fruits were studied. The results showed that compared with bagging, non-bagging cultivation increased single fruit weight and decreased fruit shape index, increased fruit sugar content, reduced acid content, and increased taste. It increased the content of vitamin C (VC) and protein in the fruit, increased the types and content of aromatic components in the fruit, significantly increased the activity of sugar- and acid-related enzymes, and improved the antioxidant capacity of the fruit. Compared to bagging cultivation, non-bagging cultivation improved the weight, taste (sugar acid ratio), and aroma of Fuji fruit, which is related to increasing the content of auxin (IAA), cytokinin (ZR), and salicylic acid (SA) and reducing the content of abscisic acid (ABA) in the fruit, as well as increasing the content of medium and trace elements calcium (Ca), iron (Fe), manganese (Mg), and boron (B). One of the mechanisms involved is the significant increase in gene expression related to phenylpropanoid biosynthesis, pentose and glucuronate interconversion, starch and sucrose metabolism, zeatin biosynthesis, microtubules, motor proteins microtubule movement, xyloglucan metabolic process, cell division, and peroxidase activity. In short, non-bagging cultivation is more conducive to improving the intrinsic quality and flavor development of Fuji apples, and one of the mechanisms is that non-bagging cultivation is beneficial for increasing the expression of related genes.

## 1. Introduction

China is the world’s largest apple-producing country, with its cultivation area and yield ranking first in the world [1]. Apple holds a pivotal position in China’s fruit industry production and is of great significance in promoting its agricultural supply-side reform, assisting rural revitalization strategy, and achieving industrial poverty alleviation and targeted poverty alleviation [2]. The Fuji series is the first main variety planted in China. The Fuji apple’s cultivation area accounts for over 70%, and its yield accounts for over 80%; Fuji apple production is important to the national economy and people’s livelihood [1,3]. Due to historical reasons such as pest control and pesticide residues [3,4,5], Fuji apples have long been cultivated through bagging. Fruit bagging has played an important role in improving appearance grade and reducing pesticide residues, and since the introduction of bagging technology from Japan in the 1990s, apple bagging has become a routine technique in flower and fruit management in China’s apple production process. As a result, China has become a major country in the world for bagging apples [4,5].

With the aging worker population and the intensification of rural labor transfer to cities, the problems brought by bagging apple cultivation have become increasingly prominent. During the bagging season, there are often “labor shortages” and “competition for people,” resulting in high production costs for fruit farmers, which have been increasing year by year. In addition, with the decline in fruit quality and exacerbation of physiological disorders in bagging cultivation [6,7], the call for non-bagging cultivation of apples is increasing, and the research on non-bagging cultivation has become a hot topic in the fields of apple breeding, cultivation, and plant protection. It is also the future of China’s apple industry. Apple non-bagging cultivation technology was identified as one of the top ten leading technologies by the Ministry of Agriculture of China in 2020 [8,9]. This means that this technology has entered the “fast lane” of promotion and application and will shoulder the heavy responsibility of promoting transformation, upgrading, improving quality and efficiency, and supporting the high-quality development of the apple industry [3].

In recent years, China has conducted many studies on the non-bagging cultivation of Fuji apples, but the research has mainly focused on suitable variety selection [10,11,12], green prevention, and control of diseases and pests in non-bagging cultivation [13,14,15]. There have been studies on differences in apple fruit quality [16,17,18], but they are limited to fruit skin coloring, total sugar and titratable acid content, and hardness, and there is no systematic and in-depth mechanism research comparing the effects of non-bagging and bagging cultivation on fruit growth and development, specific substance content changes in fruit, the characteristics of mineral absorption changes, and the accumulation trend of nutrients. Our previous research shows that compared to bagging cultivation, non-bagging cultivation changes the internal light environment of the tree canopy and improves the photosynthetic performance of the tree [19,20]. This article further shows the differences in the effects of non-bagging and bagging cultivation on specific substance content and flavor quality changes during fruit growth and development and analyzes the reasons and mechanisms in order to raise people’s understanding of the non-bagging cultivation of apple and accelerate the process of promoting non-bagging cultivation apple in China, thereby promoting the sustainable development of China’s apple industry.

## 2. Results

### 2.1. Effects of Non-Bagging and Bagging on Changes in Single Fruit Weight, Fruit Shape Index, and Diameter of Fuji Apple

As shown in Figure 1a, there was no significant difference in single fruit weight between non-bagged and bagged Fuji fruits on 30 May (before bagging) and 20 June. However, from 10 July to harvest (27 October), the single fruit weight without bagging was significantly higher than that of bagging. Figure 1b shows no significant difference in fruit shape index between non-bagged and bagged fruits on 30 May. The fruit shape index without bagging was higher than that with bagging on 20 June and 10 July, while it was lower than that with bagging from 10 September until harvest (Figure 1b). There was no significant difference in vertical diameter before bagging, from 30 May to 10 July, the longitudinal diameter of non-bagged fruits was higher than that of bagged fruits from 30 July to maturity harvest, and the vertical diameter of non-bagged fruits was lower than that of bagged fruits (Figure 1c). The effect of non-bagging on fruit horizontal diameter change was consistent with that of the single fruit weight change (Figure 1d). From 30 May to 10 July, there was no significant difference in horizontal diameter between non-bagged fruits and bagged fruits, and from 30 July to harvest, the horizontal diameter of non-bagged fruits was higher than that of bagged fruits. The effect of non-bagging on the vertical diameter and horizontal diameter was inconsistent.

### 2.2. Effects of Non-Bagging and Bagging on the Content of Hormones Related to Fuji Apple Fruit Growth and Development

The effects of non-bagging and bagging on hormone content related to fruit growth and development in the fruit of Fuji apple are shown in Table 1. The auxin (IAA) content in non-bagged and bagged fruits showed a decrease–increase trend, and the zeatin riboside (ZR) content showed an increase–decrease trend. There was no significant difference between non-bagged and bagged fruits on 30 May. And, from 20 June to harvest, these contents were higher in non-bagged fruits than in bagged fruits. The content of abscisic acid (ABA) in fruits showed a decrease–increase–decrease trend. There was no significant difference between non-bagging and bagging on 30 May. From 20 June to maturity, the content of ABA in non-bagging was lower than in bagging. From 30 May to 20 June, no salicylic acid (SA) was detected in both non-bagged and bagged fruits, and on 20 August, there was no significant difference between non-bagged and bagged fruits. In other periods, the SA content in non-bagged fruits was higher than that in bagged fruits.

### 2.3. Effects of Non-Bagging and Bagging on Sugar and Acid Accumulation of Fuji Apple Fruit

The soluble sugar content generally shows an increasing trend (Figure 2a) in both bagged and non-bagged fruit. There was no significant difference between non-bagged and bagged fruits on 30 May; however, the soluble sugar content in non-bagged fruit was higher than that in bagged fruit from 20 June to maturity. The titratable acid content generally declined (Figure 2b). There was no significant difference in titratable acid content from 30 May to 20 June between non-bagged and bagged fruits; from 10 July to harvest, that of non-bagged fruits was lower than that of bagged fruits. The sugar–acid ratio represents the taste. Figure 2c shows that the trend of the sugar–acid ratio was consistent with the trend of soluble sugars, and it was significantly higher in non-bagged fruits than in bagged fruits.

Further measurements were made on the changes in sugar component content, as shown in Figure 3. Compared with bagging, non-bagging had no significant effect on the fructose, sucrose, or glucose content of fruits before bagging treatment (30 May); from 20 June to harvest, non-bagging significantly increased the content of these sugars (Figure 3a–c), and the content of sorbitol was also significantly higher than bagging in later stages (Figure 3d).

The changes in the content of organic acid components are shown in Figure 4. Before bagging treatment (30 May), there was no significant difference in the organic acid content between the non-bagged fruit and the bagged fruit. From 20 June to harvest, the malic acid level of non-bagged fruits was lower than that of bagged fruits (Figure 4a). In the early and middle stages of fruit development, the content of citric acid in the fruit without bagging was significantly higher than that with bagging, but there was no significant difference between the two at harvest time (Figure 4b). On 20 June, the levels of succinic acid in non-bagged fruits were higher than those in bagged fruits. From 10 July to harvest, there was no significant difference between the two (Figure 4c). In the early stage of fruit development, the oxalic acid content of fruits without bagging was lower than that of fruits with bagging. In the middle stage and later stages of fruit development, the oxalic acid content of fruits without bagging was higher than that of fruits with bagging. (Figure 4d).

### 2.4. Effects of Non-Bagging and Bagging on the Activity of Sugar and Acid Metabolism Enzymes in Fuji Fruits

#### 2.4.1. Activity of Enzymes Related to Sugar Metabolism

The activity of enzymes related to sugar metabolism was measured, and the results are shown in Figure 5. From June to October, the overall sucrose phosphate synthase (SPS) activity increased, and non-bagging increased SPS activity compared to bagging (Figure 5a). The activity of sucrose synthesis direction (SS-SD) was consistent with the changes in SPS activity, and the activity of SS-SD in non-bagged fruits was significantly higher than that in bagged fruits (Figure 5b). Both the sucrose synthase decomposition direction (SS-CD) activity and acid invertase (AI) activity showed a decreasing trend, and compared to bagging, non-bagging significantly increased their activity (Figure 5c,d).

#### 2.4.2. Acid-Metabolism-Related Enzyme Activity

The determination results of fruit acid-related enzyme activity are shown in Figure 6. 

Malate dehydrogenase (NAD-MDH) activity showed a decrease–increase–decrease trend, with a relatively small amplitude. There was no significant difference in the fruit before bagging treatment. After bagging treatment, NAD-MDH activity in the fruit without bagging was significantly higher than that of bagging treatment (Figure 6a). Phosphoenolpyruvate carboxylase (PEPC) activity shows an upward–downward trend, with non-bagging increasing PEPC enzyme activity, and the effect was significant in the early stage compared to the later stage (Figure 6b). The activity of phosphoenolpyruvate carboxylation kinase (PEPCK) and malate enzyme (cyME) showed an upward trend. Overall, non-bagging had the greatest effect on PEPCK activity, followed by cyME, and had the smallest effect on NAD-MDH.

### 2.5. Effects of Non-Bagging and Bagging on Mineral Element Content of Fuji Apple Fruits 

#### 2.5.1. Major Element Contents in Apple Fruits

The impact of non-bagging and bagging on the content of macroelements in apple fruits is shown in Figure 7. From Figure 7a, it can be seen that as the Fuji apple fruit develops and matures, the nitrogen (N) content in the fruit generally shows a downward trend. Before bagging and on 30 May, there was no significant difference in N content between non-bagged and bagged fruits. On 20 August, the N content of non-bagged fruits was slightly lower than that of bagged fruits, while at other stages, the N content of non-bagged fruits was lower than that of bagged fruits. The change in phosphorus (P) content in fruits showed a trend of decrease–increase–decrease. The P content of non-bagged fruits was significantly lower than that of bagged fruits (Figure 7b). The change in potassium (K) content in fruits also showed a trend of decrease–increase–decrease. There was no significant difference in K content between non-bagged and bagged fruits on 30 May, 20 October, and 27 October; however, the K content of non-bagged fruits was lower than that of bagged fruits in other periods (Figure 7c).

#### 2.5.2. Medium and Trace Element Contents in Fruits of Apple

The effects of non-bagging and bagging on the content of trace elements in fruits are shown in Figure 8.

With the extension of the apple growth period, the overall calcium (Ca) content in the fruit shows a trend of increasing–decreasing–increasing–decreasing. The value reached its lowest on 20 August and reached its peak on 30 September, which was assumed to be related to the weather temperature at that time. On 30 May, there was no significant difference in calcium content between non-bagged and bagged fruits. From 20 June to maturity, the calcium content of non-bagged fruits was significantly higher than that of bagged fruits. From June to October, compared to bagging, the Ca content in non-bagged fruits increased by 0.87 to 3.58 times, respectively (Figure 8a).

The iron (Fe) content of fruits showed an upward–downward trend, and after 10 July, the Fe content of non-bagged fruits was significantly higher than that of bagged fruits (Figure 8b).

The magnesium (Mg) content in the fruits showed a downward–upward–downward trend. On 20 June, the Mg content of non-bagged fruits was higher than that of bagged fruits. However, from 10 July to 20 August, there was no significant difference in Mg content between non-bagged and bagged fruits. From 20 August to maturity, the Mg content of non-bagged fruits was significantly higher than that of bagged fruits (Figure 8c).

The manganese (Mn) content showed a decreasing–increasing–decreasing trend, and from 20 June to maturity, the Mn content of non-bagged fruits was significantly higher than that of bagged fruits (Figure 8d).

The content of boron (B) in fruits showed an increasing–decreasing trend. After 30 July, the content of B in fruits tended to stabilize, and the content of B in non-bagged fruits was significantly higher than that in bagged fruits. From July to October, the content of fruit B without bagging increased by 0.58 to 15.99 times compared to bagging, respectively (Figure 8e).

The zinc (Zn) content in fruits shows a trend of increasing–decreasing–increasing–decreasing. There was no significant difference in Zn content between non-bagged and bagged fruits on 30 May and 20 June. From 10 July to 30 September, the zinc content of non-bagged fruits was higher than that of bagged fruits, while on 20 October and 27 October, the zinc content of non-bagged fruits was lower than that of bagged fruits (Figure 8f).

Non-bagging increased Ca content the most significantly, followed by Fe, Mn, and B.

### 2.6. Effects of Non-Bagging and Bagging on Changes in Vitamin C (VC) and Soluble Protein Content of Fruits

Before bagging treatment (30 May), there was no significant difference in vitamin C (VC) content between non-bagged and bagged fruits. On 20 June, the VC content of non-bagged fruits was lower than that of bagged fruits. From 10 July to maturity, the VC content of non-bagged fruits was significantly higher than that of bagged fruits (Figure 9a).

Before bagging (30 May) and 30 July, there was no significant difference in soluble protein content between non-bagged fruits and bagged fruits. In other periods, the soluble protein content of non-bagged fruits was significantly higher than that of bagged fruits (Figure 9b).

### 2.7. Effects of Non-Bagging and Bagging on the Aroma Substances of Fuji Apple Fruits during Harvesting Period

The total ion flow diagrams of aroma components in Fuji apples with and without bagging are shown in Figure 10. There are certain differences in the retention time and relative content of aroma components between non-bagging and bagging treatments.

Compared to bagging, non-bagging increased the variety and content of fruit aroma substances (Figure 10 and Table 2).

From Figure 10, it can be seen that there are certain differences in the retention time and relative content of aroma components between non-bagged and bagged fruits. After search and analysis, total alcohols, aldehydes, esters, ketones, and other compounds were detected, and their contents are shown in Table 2. There are more types of aromatic substances in Fuji apples in non-bagged than bagged fruits. Thirty-three types were detected without bagging, while twenty-one types were detected with bagging. From Table 2, it can be seen that the categories with higher relative content without bagging and bagging are aldehydes, alcohols, and esters, with 26.54%, 48.01%, and 13.99% of non-bagged fruits, respectively. The bagged fruits were 47.37%, 46.60%, and 2.55%, respectively. The alcohol content of non-bagged fruits was lower than that of bagged fruits, while the ester content was higher than that of bagged fruits, and the difference in aldehyde content was not significant.

### 2.8. Effects of Non-Bagging and Bagging on Malondialdehyde (MDA) Content and Antioxidant Enzyme Activity in Fuji Apple Fruits

As shown in Figure 11a, during the fruit development period, the malondialdehyde (MDA) content of fruit showed a general upward trend. Compared with 30 May, the MDA content of fruit of the bagging treatment on 20 June significantly increased, indicating that bagging treatment caused oxidative stress. The MDA content of fruits without bagging during the entire growth period was significantly lower than that of bagging treatment except for 30 May.

Further, the main antioxidant enzyme activities were measured as shown in Figure 11b–d. The trend of superoxide dismutase (SOD) activity in non-bagged and bagged fruits was similar, showing an increasing–decreasing–increasing–decreasing trend. The peaks appeared on 30 July and 30 September, respectively. Throughout the growth period, non-bagged fruits were consistently higher than bagged fruits, especially from 30 July to 30 September (Figure 11b). The peroxidase (POD) activity showed an increasing–decreasing trend, with no significant difference except before bagging (30 May). The POD activity of non-bagged fruits was significantly higher than that of bagged fruits throughout the entire growth period (Figure 11c). The trends of catalase (CAT) activity and POD activity changes are similar, with no bagging higher than bagging fruit CAT activity. From Figure 11, it can also be seen that non-bagging improved fruit antioxidant enzyme activity, with the greatest impact on CAT, followed by POD and SOD.

### 2.9. Effects of Non-Bagging and Bagging on Gene Transcription and Expression in Fuji Apple Fruits

Figure 12a shows the changes in gene expression in fruit peels treated with and without bagging, with 1571 differentially expressed genes, of which 1269 were upregulated and 302 were downregulated.

We conducted Kyoto Encyclopedia of Genes and Genomes (KEGG) pathway enrichment analysis on upregulated genes, as shown in Figure 12b. The results showed that most of the upregulated genes were enriched in metabolic pathways, such as phenylpropanoid biosynthesis, pentose and glucuronate interconversion, starch and sucrose metabolism, and zeatin biosynthesis.

A GO (gene ontology) functional enrichment analysis was performed on upregulated genes, as shown in Figure 12b. From the figure, it can be seen that upregulated genes are enriched in biological processes, cellular components, and molecular functions, and the number of enriched genes was not significantly different. Therefore, the number of enriched differential genes in the functional classification of cellular components is slightly higher, and most of them are concentrated in extracellular regions, microtubules, motor proteins, etc. In biological processes, upregulated genes are mainly enriched in microtubule movement, xyloglucan metabolic process, and cell division, while in molecular function, they are mainly enriched in microtubule motor activity, microtubule binding, peroxidase activity, etc.

## 3. Discussion

### 3.1. Improving Single Fruit Weight and Reducing Fruit Shape Index of Fuji Apple by Non-Bagging Cultivation

From 10 July to harvest, the single fruit weight of non-bagged fruits was significantly higher than that of bagged fruits, which is consistent with previous research results [21,22]. This is because bagging creates a high-temperature and -humidity microenvironment for the fruit [23], which affects its development. Meanwhile, the closed environment of the fruit bag can also inhibit the synthesis of chlorophyll on the fruit skin, which directly affects the synthesis and accumulation of nutrients in the fruit itself, leading to the bagged fruit becoming smaller. This is consistent with the research results on sugar, protein, etc., in this article (Figure 3 and Figure 7), indicating that non-bagging may be beneficial for improving yield, thus improving the economic benefits of fruit farmers.

The impact of non-bagging cultivation on the Fuji fruit shape index is relatively complex. In the early stage of fruit development, the fruit shape index of non-bagging was higher than bagging, and in the later stage, it was lower than bagging. Non-bagging cultivation reduced the fruit shape index at the mature stage, which was consistent with Yue et al.’s research results [22]. However, there are also opposite results [24,25]. Some studies also suggested that non-bagging has no significant impact on fruit shape index [26,27], which may be due to differences in the types of fruit bags used, bagging time, variety types, and climate conditions.

The impact of non-bagging on fruit shape index may also be related to the different effects of different stages on fruit growth-related hormones (Table 1). In the early stage of fruit development (20 June), the IAA content in non-bagged fruits was significantly higher than that in bagged fruits, while there was no significant difference in ZR content between the two. On 20 June, the non-bagging of fruit produced higher IAA than that of ZR; however, from 30 July to 10 September, the effect of non-bagging on increasing IAA content in fruits was lower than ZR content. This changed again from 30 September to harvest, whereas the effect of non-bagging on increasing IAA content was higher than ZR. The impact of non-bagging on hormone content may be one of the reasons why non-bagging increases single fruit weight while reducing the ripening fruit shape index. According to our previous research results, the increase in single fruit weight without bagging is related to the improvement of tree photosynthetic capacity, which is beneficial for water transportation [20].

### 3.2. Non-Bagging Cultivation to Increase Sugar Content, Reduce Acid Content, and Improve Fruit Taste of Fuji Apples

Compared with bagging, non-bagging significantly increased the soluble sugar content of the fruit, decreased the titratable acid content, and increased the taste (sugar–acid ratio) of the fruit (Figure 2). The determination results of sugar acid components further confirmed this conclusion (Figure 3 and Figure 4).

The sugar content of non-bagged fruits was higher than that of bagged fruits, which is consistent with previous research results [28,29]. This is related to the decrease in the photosynthetic rate of apple leaves and fruits due to the blocking effect of the fruit bag [20,30]. At the same time, bagging forms a greenhouse effect on the fruit [23], which increases the respiratory intensity of the fruit in high-temperature environments and increases the consumption of carbohydrates, which is not conducive to the accumulation of sugar. During the mature period, compared to bagging, the contents of fructose, sucrose, and sorbitol in fruits without bagging significantly increased (Figure 3). It is speculated that non-bagging cultivation may significantly improve the accumulation and transportation of photosynthetic products in Fuji apples. Compared with bagging, non-bagged fruits significantly reduced titratable acid content. This is consistent with the research of Xue et al. [31] on Huaniu apples and Yang et al. [18] on Fuji apples, but there are also opposite conclusions [28,32]. This may be related to the inconsistency of the variety of the research object. The increase in fruit sugar–acid ratio without bagging was not only related to sugar increase but also to acid reduction, both of which were related to the increase in enzyme activity related to fruit sugar acid metabolism without bagging (Figure 4 and Figure 5). Some studies have proved that cytokines are involved in regulating extracellular invertase and hexose transport protein and promoting carbohydrate transport to fruits [33,34]. Our research results indicated that compared to bagging, non-bagging increased the levels of endogenous hormones IAA, ZR, and SA in fruits (Table 1). The influence of non-bagging cultivation methods on hormone levels may also be a mechanism for increasing sugar and reducing acid. Further research is needed on the mechanism.

### 3.3. Cultivation without Bagging to Improve the Nutritional Composition and Aroma Substances of Fuji Apple Fruit

Compared with bagging, non-bagging significantly increased the VC content and soluble protein content in Fuji fruits (Figure 9), which is beneficial for improving fruit quality. VC is an important nutrient and a strong antioxidant, playing an important role in metabolic regulation [35]. VC is transferred to the fruits for storage after being synthesized by the carbon anabolism of the leaves. The high content of VC in fruits without bagging may be related to the fact that fruits are exposed to the natural environment and the automatic defense mechanism starts to accelerate the transfer of VC in leaves to the fruits, or it may be related to the increase in key enzyme activities of VC synthesis and metabolism in fruits without bagging [36]. The specific mechanism needs further research. Fruit aroma is one of the important factors in the formation of flavor quality. Compared with bagging, non-bagging increases the types of individual characteristic aroma components in apples and increases the content of fruit characteristic aroma components (*p* < 0.05) (Figure 10 and Table 2). This is consistent with previous research findings on ‘Golden Delicious’, ‘Qin Guan’, and ‘Ruixue’ [37,38]. Non-bagging cultivation can increase the activity of sugar and acid metabolism enzymes, which may be related to the key enzyme activity of enhancing aroma components and enhancing aroma substance synthesis without bagging [39]; the texture and structure of fruits play a crucial role in the release of apple aroma [40]. This study found that there is a significant difference in gene expression between non-bagging and bagging fruit skins, especially when non-bagging enhances genes related to fruit skin structure. It is speculated that the increase in aroma substances may also be related to the increase in gene expression related to aroma substance synthesis without bagging (Figure 12).

### 3.4. Different Effects of Non-Bagging and Bagging Cultivation on the Absorption and Accumulation of Mineral Nutrition in Fuji Apple Fruits

Compared to bagging, non-bagging reduces the number of elements in the fruit while increasing the number of trace elements, especially Ca, Fe, and B.

It is generally believed that N, P, and K have strong mobility in plants and mainly exist in the tissues and organs with the most active metabolism. The content of N, P, and K in non-bagged fruits is lower than that in bagged fruits. This is mainly due to the higher temperature inside the bag compared to the outside, active fruit metabolism, and the transfer of N, P, and K from plant leaves to the fruit after bagging. He et al. also obtained similar results [41] with significantly lower differences compared to this article. Non-bagging increased the calcium content of fruits (Figure 8a), which is consistent with the research results of Geng Jun et al. [16]. The absorption of Ca in the fruit is mainly passive. After bagging Fuji apples, the water transpiration of the fruit decreases, and the amount of Ca ions flowing into the fruit with transpiration also decreases. Therefore, the Ca content of the fruit is lower than that of non-bagged fruits. The content of B in non-bagged fruits is higher than that in bagged fruits, especially in the later stage of fruit development. This is because B and Ca are transported by extracellular vesicles, and the absorption of B by leaves and fruits is in a mutually beneficial relationship with the absorption of Ca, which is consistent with the experimental results of Niu et al. [42]. The content of Fe and Mn in non-bagged fruits is also significantly higher than that in bagged fruits. According to the antagonistic relationship of mineral elements, it seems to be contradictory due to the effect of high B on the absorption of Fe and the reduction in Fe content in plants. Ca and Fe hinder the absorption of Mn, and it is also contradictory because the content of Mn in non-bagged fruits is higher than that in bagged fruits. It is speculated that their content has not reached the antagonistic level, but the reasons need further study. This study also found that compared to non-bagged fruits, bagged fruits have higher N content and lower Ca content, resulting in higher N/Ca, which may increase the occurrence of physiological disorders such as fruit brown spot, bitter pox, and tiger skin disease [43,44].

### 3.5. Improving the Antioxidant Capacity of Fuji Apple Fruits by Non-Bagging Cultivation

As the fruit matures and ages, the fruit’s oxidative function will continue to increase, leading to the accumulation of reactive oxygen species. Excessive reactive oxygen species will damage the membrane system, causing the degradation and peroxidation of membrane lipids, producing the peroxide product MDA, and leading to cell membrane leakage and protein denaturation [45]. The MDA content can indirectly indicate the degree of plant damage [46]. In this study, the MDA content showed an upward trend throughout the entire growth and development of apples, and the MDA content in non-bagged fruits was lower than that in bagged fruits throughout the entire growth and development process. The activities of SOD, POD, and CAT in the fruits were significantly increased, which is similar to previous research results [47,48]. It can be seen that non-bagging cultivation increases the activity of protective enzymes such as SOD, POD, and CAT in apple fruits, reduces the production of reactive oxygen species, weakens membrane lipid peroxidation reaction, and accumulates MDA at a lower rate than the bagging cultivation. This also fully proves that non-bagging fruits have stronger antioxidant capacity than bagging fruits. Non-bagging cultivation activates the self-protection mechanism of fruit, which is also one of the reasons for the accumulation of certain contents in fruits.

### 3.6. The Gene Expression of Fuji Apple Fruit between Non-Bagging and Bagging Cultivation

Considering the significant differences in fruit quality between bagged and non-bagged apples, further analysis was conducted on gene expression between bagged and non-bagged apples. The study found a total of 1571 differentially expressed genes, of which 1269 were upregulated, significantly higher than the 302 downregulated genes (Figure 12a). Further analysis revealed that upregulated genes were mainly enriched in processes such as glucose metabolism, zeatin synthesis, microtubules, motor proteins, cell division, and POD activity. This is consistent with our measurements of sugar and acid content, hormone content, and antioxidant activity (Figure 3 and Figure 11, Table 1), indicating that the increase in fruit weight, sugar content, and antioxidant capacity is related to the upregulation of related gene expression without bagging.

## 4. Materials and Methods

### 4.1. Basic Information of Test Site

The experiment was conducted in the Taidong base of the Shandong Institute of Pomology (36°11′07″ N, 117°06′51″ E) in 2022. The area belongs to a temperate continental semi-humid monsoon climate. In 2022, the annual rainfall was 631 mm, the average temperature was 15 °C, and there was an extreme high temperature of 39 °C (21 June) and an extreme low temperature of −11 °C (16 December) (https://www.tianqi24.com/taian1/history2022.html (accessed on 1 March 2023).

The orchard is a plain orchard with sandy loam soil and medium cultivation and management levels. The tested variety was a thirteen-year-old Red Fuji apple (Tianhong.2/SH/*M. robusta*), with north–south directionality and a plant row spacing of 1.5 m × 3 m.

Conventional fertilization and watering were used. In late November of 2022, sufficient base fertilizer (8325 kg/km^2^) was applied, urea (1665 kg/km^2^ in late March) was applied before flowering (early April), and N, P, and K compound fertilizers (832.5 kg/km^2^) were applied in late June. Flowers and fruits were thinned manually at flowering and obtained young fruit stages according to a load of 60,000 kg/km^2^ (about 140 fruits/tree).

For the experiment, plants with the same tree strength and uniform fruit-bearing capacity (about 140 fruits/tree) were selected, and the bagging treatment was carried out using double-layer paper bags (Japan, KM-2, red on the inside and brown on the outside). They were unified on 2 June, with the picking of bags on 11 October uniformly and harvesting on 27 October. Conventional pest control mainly included the control of aphids in late April and the spraying of insecticides and fungicides once before bagging (late May).

### 4.2. Test Design

There were two treatments in the experiment, which were recorded as non-bagging and bagging, with three replicates for each treatment and one replicate for every 5 trees at 15 trees per treatment. Starting from 30 May (before bagging), samples were taken every 20 days until maturity and harvest (27 October). The apple maturity stage was determined according to the flavor, color, hardness, and other indicators of the apples. A total of nine samples were taken, ensuring the weight and diameter of a single fruit. We simultaneously took some fruits, froze them quickly in liquid nitrogen, and stored them at −80 °C until further analysis.

### 4.3. Measuring Methods

#### 4.3.1. Fruit Size and Shape Index

According to a method described by zhang [49], we used an electronic balance to measure the weight of a single fruit, measured the vertical and horizontal diameter of the fruit using a vernier caliper, and calculated the fruit shape index. Fruit shape index = vertical diameter/horizontal diameter.

#### 4.3.2. Determination of Hormone Level

The levels of IAA, ZR, ABA, and SA in the Fuji apples were determined through high-performance liquid chromatography (HPLC) according to a method described by Almeida Trapp et al. [50].

Samples were weighed to 0.1−0.2 g, ground with liquid nitrogen, and extracted with acetonitrile. This was repeated three times, combining the three extracts, with the extracts centrifuged and concentrated until dry. They were then dissolved in phosphate buffer, with pigments removed, extracted with ethyl acetate, centrifuged until dry, dissolved in methanol, and passed through 0.22 μm organic phase membrane filtration. The HPLC system was equipped with an SPD-20A UV/Vis detector (Shimadzu, Kyoto, Japan) and a Venusil XBP C18 column (100 mm × 4.6 mm).

#### 4.3.3. Determination of Sugar and Acid Content of Fruit

The content of soluble sugar was determined using the anthrone method [51]. Titratable acid content was determined by using acid–base neutralization titration [52].

HPLC was used for the determination of sugar and acid components [53,54]. Samples were weighed to 1–2 g, ground under liquid nitrogen conditions, transferred to a centrifuge tube, and mixed with the extraction reagent. Ultrasonic extraction was performed, and the extraction process was repeated three times for each sample. The centrifugal supernatant was recovered at a constant volume, filtered with a filter membrane, and injected into the instrument for analysis.

Chromatographic conditions for the determination of free sugar were as follows: Shimadzu differential detector; acetonitrile: water as mobile phase; 30 °C; amino column; 1 mL/min flow rate for detection; and 20 μL injection volume. Chromatographic conditions for the determination of organic acids were as follows: Shimadzu liquid phase UV detector; phosphate solution and methanol as mobile phase; C18 chromatographic column to detect at a certain gradient mobile phase; 30 °C; 0.4 mL/min flow rate; 210 nm wavelength; and 10 μL injection volume. The contents of the sugar and acid components were calculated according to the peak area of the sample and the standard curve.

#### 4.3.4. Determination of Sugar-Metabolism-Related Enzyme Activity in Fruit

The enzyme solution was extracted following Lowell et al. [55] and Wang et al. [56]. The dialyzed enzyme solution was used in the determination of the activities of SPS, SS, and AI. SPS and AI activity determination was performed using the methods of Zhu, Komor, and Moore [57]. The SS-CD assay was based on Huber’s method [58], and the SS-SD assay was based on the methods of Lowell et al. [55] and Hubbard et al. [59].

#### 4.3.5. Determination of Enzyme Activity Related to Acid Metabolism in Fruit

The enzyme solution preparation of the acid-metabolism-related enzymes NAD-MDH, cyME, and PEPCK referred to the methods of Hirai et al. [60]. NAD-MDH, cyME, and PEPCK activities were determined using methods of Hirai and Luo Ancai [60,61].

#### 4.3.6. Determination of Mineral Element Content of Fruit

The extraction and quantification of mineral elements in fruits refer to Lu et al. [62]. Fruits are crushed and then boiled with sulfuric acid–perchloric acid. The contents of N were determined by using the semi-micro Kjeldahl method; the contents of P, K, Ca, Mg, B, Fe, Mn, and Zn were determined by using an inductively coupled plasma emission spectrometer (ICP-AS) (OPTIMA3300DV, Perkin Elmer, Shelton, CT, USA). The argon flow rate was 15 L/min, and the injection volume was 1.5 mL/min.

#### 4.3.7. Determination of Other Main Substances Contents of Fruit

VC was measured through a 2, 6-dichloroindophenol sodium titration [63].

Soluble protein content was determined by using the Coomassie brilliant blue G250 dyeing method according to the Bradford method [64].

#### 4.3.8. Determination of Volatile Matter Content in Apple Fruits

The aroma quality of fruits was determined according to the method of Wang A. R. et al. [65]. The determination of fruit aroma quality uses headspace solid-phase extraction (SPME) technology to extract the aroma components of the fruit, determined and analyzed by using gas chromatography–mass spectrometry (GCMS-QP2010, Shimadzu, Kyoto, Japan).

#### 4.3.9. Determination of Sugar-Metabolism-Related Enzyme Activity in Fruit

The MDA level was assayed according to Quan et al. [66].

Total SOD, CAT, and POD activities were determined as described by Prochazkova et al. [67]. A total of 0.5 g of leaves was placed in a pre-cooled mortar, and 5 mL of pre-cooled phosphate buffer was added to the ice bath. The leaves were ground and centrifuged at 12,000× *g* at 4 °C for 10 min. The supernatants were collected and used to assay antioxidative enzymatic activities. Results were expressed as enzyme activity based on unit fresh weight (FW).

#### 4.3.10. Gene Expression Analysis

Total RNA was extracted using the TRIzol reagent (Invitrogen, Carlsbad, CA, USA) according to the manufacturer’s protocol. RNA purity and quantification were evaluated using the NanoDrop 2000 spectrophotometer (Thermo Scientific, Waltham, MA, USA). RNA integrity was assessed using the Agilent 2100 Bioanalyzer (Agilent Technologies, Santa Clara, CA, USA). Then, the libraries were constructed using the VAHTS Universal V6 RNA-seq Library Prep Kit according to the manufacturer’s instructions. The transcriptome sequencing and analysis were conducted by OE Biotech Co., Ltd. (Shanghai, China). Double-ended sequencing was performed using Illumina Novaseq 6000 (Illumina, San Diego, CA, USA) to obtain sequencing data.

After the quality evaluation of sequencing data, quality control data were obtained and compared with the apple reference genome on HISAT2 software [68,69] to obtain sample genes. Differential expression analysis was conducted using DESeq2 software, and gene functional annotation analysis was conducted using GO and KEGG databases to analyze the metabolic pathways involved in differential genes.

### 4.4. Statistical Analysis

All experiments were conducted in triplicate, and the presented values represent the mean ± standard error (S.E.) of three replicates. The test results were plotted with Sigmaplot 10.0, and the statistical analysis was conducted using the Data Processing System (DPS; Zhejiang University, Hangzhou, China). Differences among the treatments or periods of the same treatment were compared using Duncan’s multiple range tests at 0.05 probability levels (*p* ≤ 0.05).

## 5. Conclusions

In summary, compared with bagging cultivation, non-bagging cultivation improved the weight, taste (sugar–acid ratio), and aroma of Fuji fruit, which is related to increasing the content of IAA, ZR, and SA and reducing the content of ABA in the fruit, as well as increasing the content of Ca, Fe, Mg, and B. One of the mechanisms involved is the significant increase in gene expression related to phenylpropanoid biosynthesis, pentose and glucuronate interconversion, starch and sucrose metabolism, zeatin biosynthesis, microtubule movement, motor proteins, xyloglucan metabolic process, cell division, and peroxidase activity.

## Figures and Tables

**Figure 1 plants-12-03309-f001:**
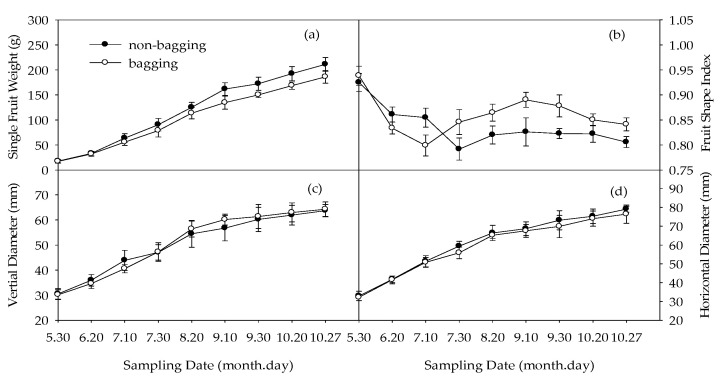
Effects of non-bagging and bagging on changes in single fruit weight (**a**), fruit shape index (**b**), vertical diameter (**c**), and horizontal diameter (**d**) of Fuji fruit. The values are mean ± standard error (S.E.) of three replicates. Bars represent S.E. Fruit shape index = vertical diameter/horizontal diameter.

**Figure 2 plants-12-03309-f002:**
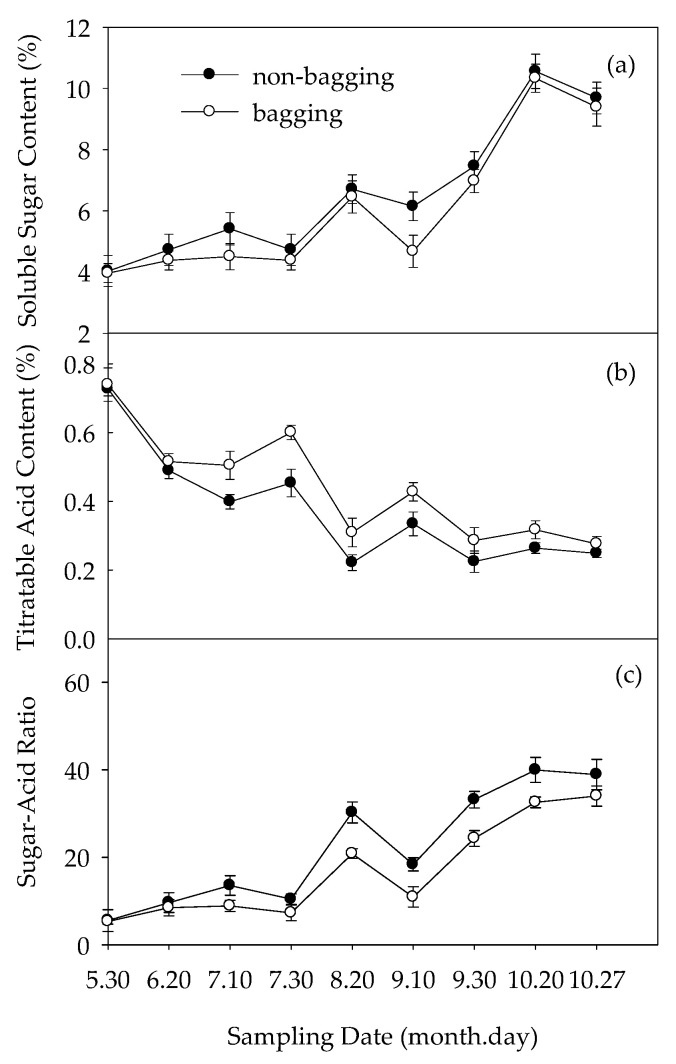
Effects of non-bagging and bagging on soluble sugar content (**a**), titratable acid content (**b**), and sugar–acid ratio (**c**) of fruit; the values are mean ± standard error (S.E.) of three replicates. Bars represent S.E.

**Figure 3 plants-12-03309-f003:**
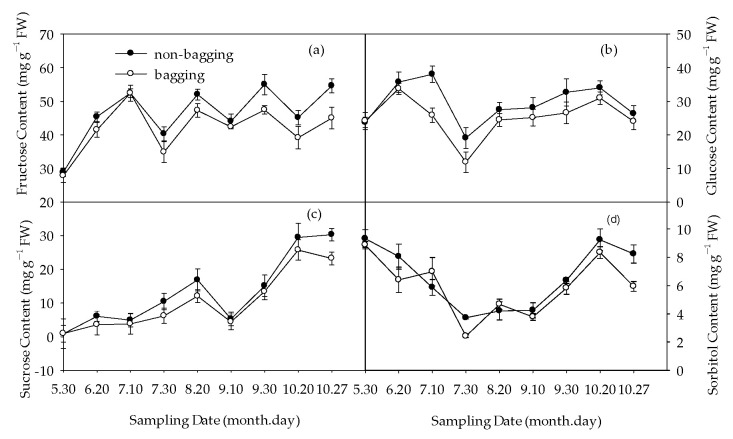
Effects of non-bagging and bagging on the accumulation of fructose (**a**), glucose (**b**), sucrose (**c**), and sorbitol (**d**) in apple fruit. The values are mean ± standard error (S.E.) of three replicates. Bars represent S.E. FW refers to fresh weight; the same below.

**Figure 4 plants-12-03309-f004:**
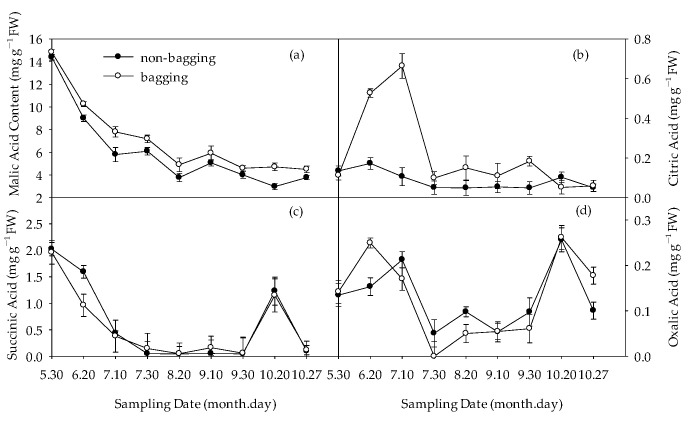
Effects of non-bagging and bagging on malic acid (**a**), citric acid (**b**), succinic acid (**c**), and oxalic acid (**d**) contents in apple fruit. The values represent mean ± standard error (S.E.) of three replicates. Bars indicate S.E.

**Figure 5 plants-12-03309-f005:**
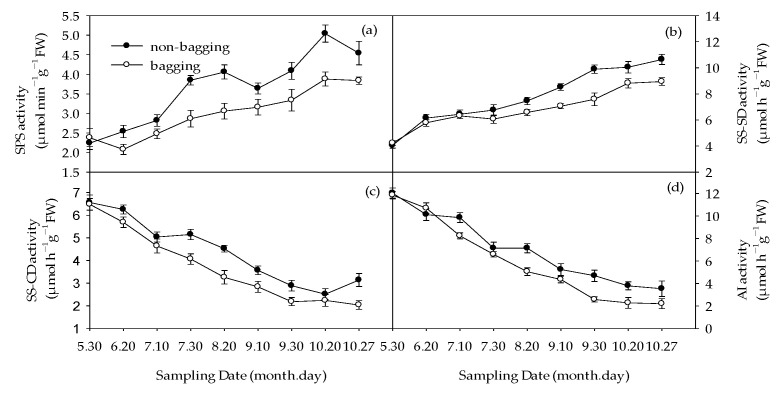
Effects of non-bagging and bagging on sucrose phosphate synthase (SPS) activity (**a**), sucrose synthesis direction (SS-SD) activity (**b**), sucrose synthase decomposition direction (SS-CD) activity (**c**), acid invertase (AI) activity (**d**) on apple fruit. The values represent mean ± standard error (S.E.) of three replicates. Bars indicate S.E.

**Figure 6 plants-12-03309-f006:**
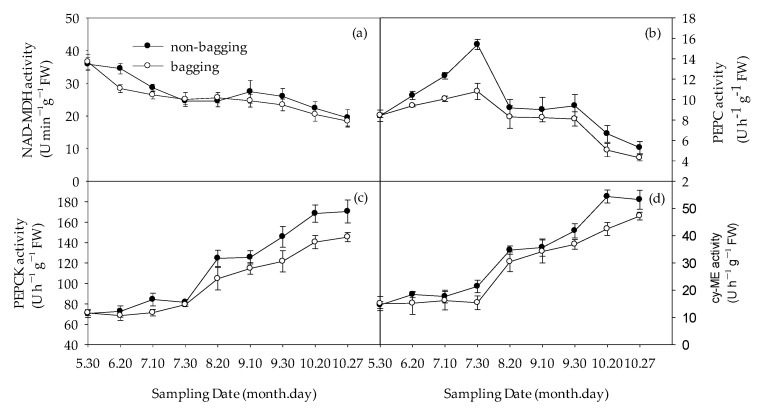
Effects of non-bagging and bagging on Malate dehydrogenase (NAD-MDH) activity (**a**), Phosphoenolpyruvate carboxylase (PEPC) activity (**b**), phosphoenolpyruvate carboxylation kinase (PEPCK) activity (**c**), and malate enzyme (cyME) activity (**d**) of Fuji apple fruit. The values represent mean ± standard error (S.E.) of three replicates. Bars indicate S.E.

**Figure 7 plants-12-03309-f007:**
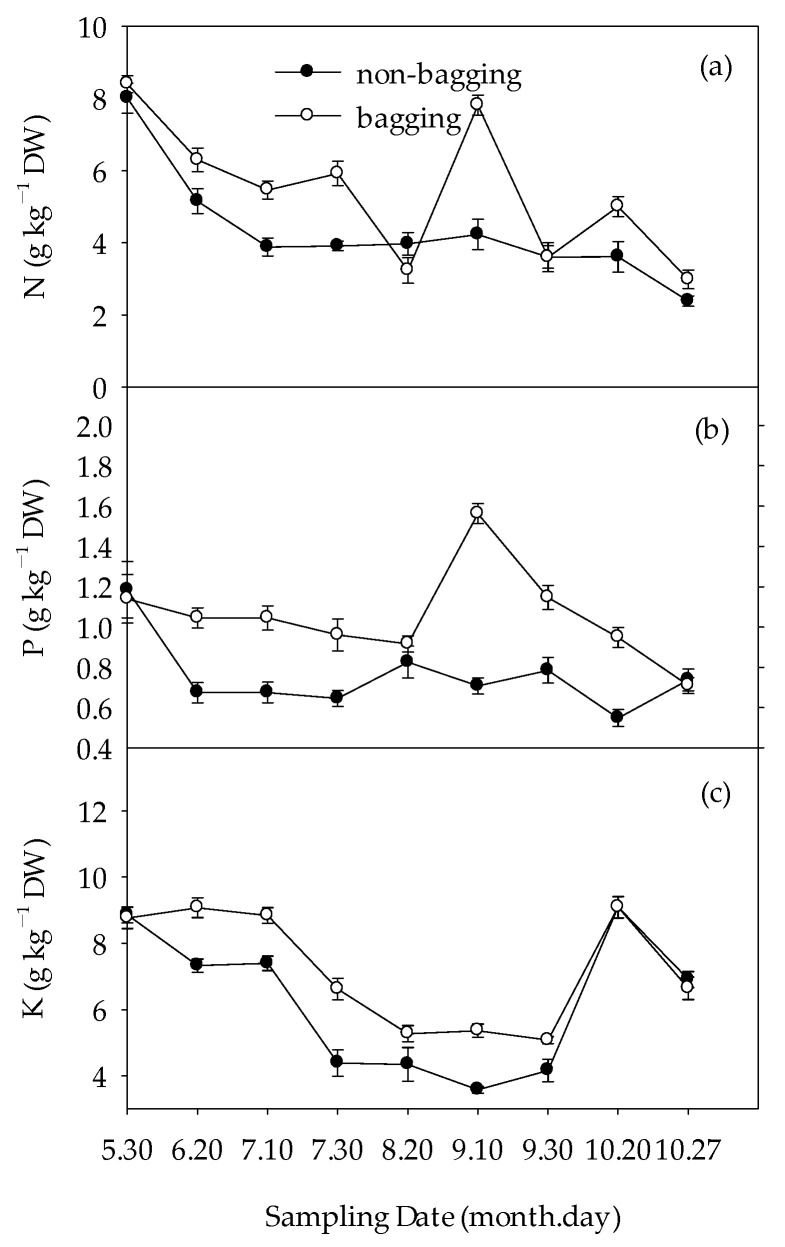
Effects of non-bagging and bagging on macroelements nitrogen (N) (**a**), phosphorus (P) (**b**), and potassium (K) (**c**) content in apple fruit. The values represent mean ± standard error (S.E.) of three replicates. Bars indicate S.E. Bars with the same letter were not significantly different at *p* < 0.05. DW refers to dry weight; the same below.

**Figure 8 plants-12-03309-f008:**
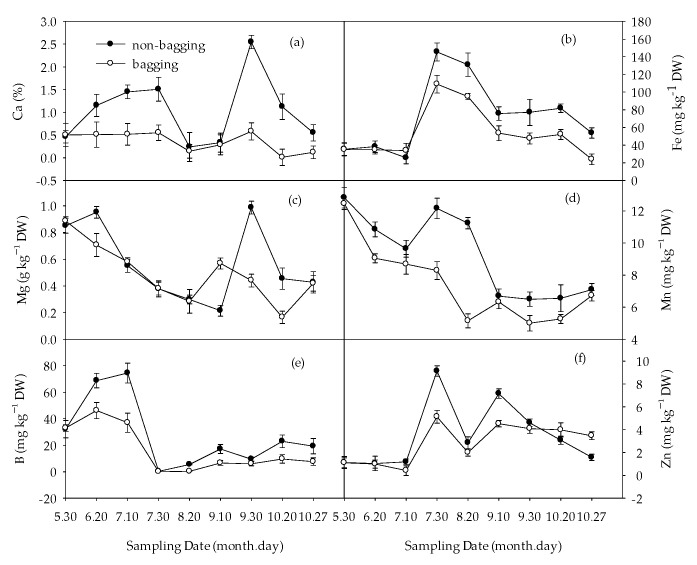
Effects of non-bagging and bagging on medium and trace elements calcium (Ca) (**a**), iron (Fe) (**b**), magnesium (Mg) (**c**), manganese (Mn) (**d**), boron (B) (**e**), and zinc (Zn) (**f**) content in apple fruit. The values represent mean ± standard error (S.E.) of three replicates. Bars indicate S.E.

**Figure 9 plants-12-03309-f009:**
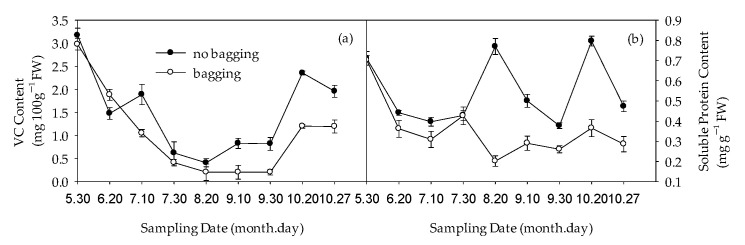
Effects of non-bagging and bagging on Vitamin C (VC) (**a**) and soluble protein (**b**) content of fruit. The values represent mean ± standard error (S.E.) of three replicates. Bars indicate S.E.

**Figure 10 plants-12-03309-f010:**
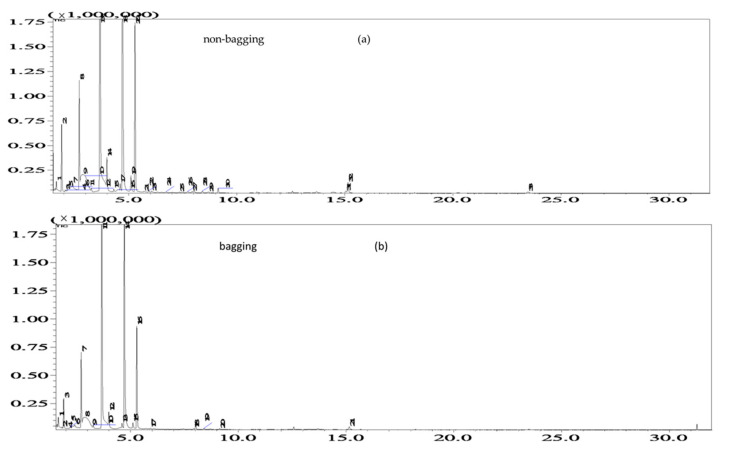
Total ion chromatograms of volatile constituents in non-bagging (**a**) and bagging apple fruits (**b**) of headspace solid-phase microextraction (HS-SPME). Note: sampling date was 27 October.

**Figure 11 plants-12-03309-f011:**
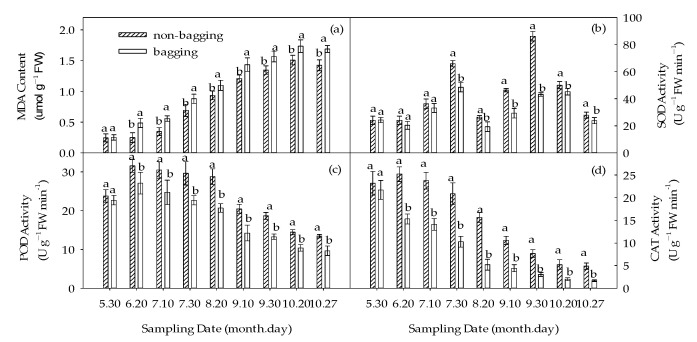
Effects of non-bagging and bagging on malondialdehyde (MDA) content (**a**), superoxide dismutase (SOD) activity (**b**), peroxidase (POD) activity (**c**), and catalase (CAT) activity (**d**) of fruit. The values represent mean ± standard error (S.E.) of three replicates. Bars indicate S.E. The same lowercase letters above bars indicate no significant difference among treatments (*p* > 0.05).

**Figure 12 plants-12-03309-f012:**
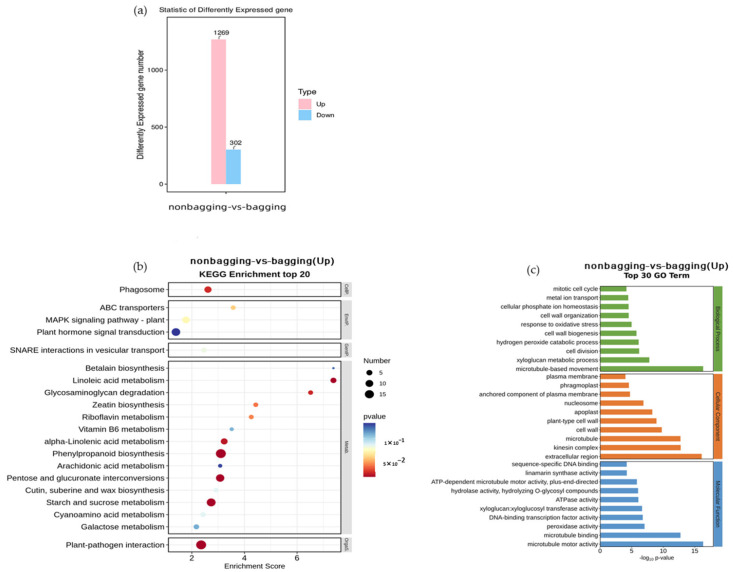
Statistical analysis of differentially expressed genes (**a**); Kyoto Encyclopedia of Genes and Genomes (KEGG) (**b**) and gene ontology (GO) (**c**) enrichment analysis of upregulated genes.

**Table 1 plants-12-03309-t001:** Effect of non-bagging and bagging on the content of hormones related to fruit growth and development in Fuji apple.

Sampling Date (Month.Day)	Treatment		Hormone Content (μg g^−1^ FW)		
		IAA	ZR	ABA	SA
5.30	non-bagging	23.00 ± 1.6 a	27.18 ± 1.01 a	0.18 ± 0.005 a	-
	bagging	22.89 ± 1.1 a	26.46 ± 2.03 a	0.17 ± 0.004 a	-
6.20	non-bagging	13.07 ± 0.61 a	7.27 ± 1.02 a	0.06 ± 0.005 b	-
	bagging	5.47 ± 0.64 b	6.73 ± 1.56 ab	0.09 ± 0.005 a	-
7.30	non-bagging	0.32 ± 0.07 a	0.22 ± 0.09 a	0.02 ± 0.007 ab	1.02 ± 0.05 a
	bagging	0.23 ± 0.03 ab	0.00 ±0.00 b	0.03 ± 0.003 a	0.84 ± 0.04 b
8.20	non-bagging	0.14 ± 0.05 a	0.35 ± 0.05 a	0.04 ± 0.004 b	0.68 ± 0.03 a
	bagging	0.03 ± 0.02 b	0.09 ± 0.01 b	0.11 ± 0.006 a	0.61 ± 0.03 a
9.10	non-bagging	0.09 ± 0.05 a	1.18 ± 0.05 a	0.04 ± 0.006 b	1.00 ± 0.04 a
	bagging	0.04 ± 0.03 ab	0.18 ± 0.03 b	0.07 ± 0.004 a	0.86 ± 0.05 b
9.30	non-bagging	0.76 ± 0.07 a	0.17 ± 0.06 a	0.03 ± 0.002 a	0.99 ± 0.06 a
	bagging	0.38 ± 0.04 b	0.15 ± 0.04 ab	0.03 ± 0.006 a	0.43 ± 0.07 b
10.27	non-bagging	1.40 ± 0.09 a	0.17 ± 0.05 a	0.05 ± 0.003 b	0.79 ± 0.09 a
	bagging	0.48 ± 0.07 b	0.12 ± 0.02 b	0.10 ± 0.004 a	0.44 ± 0.04 b

Note: Different lowercase letters after the same column of data indicate that the difference between treatments was significant at the level of *p* ≤ 0.05; “-” indicates no significant difference detected; FW refers to fresh weight.

**Table 2 plants-12-03309-t002:** Gas chromatography/mass spectrometry (GC/MS) analysis result of the volatile constituents in non-bagged and bagged Fuji apple ripe fruits.

Categories	Compounds	Relative Content (%)	
		Non-Bagging	Bagging
Alcohols	1-Propanol, 2-methyl-	0.44 ± 0.05 b	1.02 ± 0.02 a
	1-Butanol	3.13 ± 0.1 a	0.05 ± 0.01 b
	1-Penten-3-ol	0.09 ± 0.01 a	0.05 ± 0.01 b
	2-Methyl-1-butanol	5.77 ± 1.01 a	6.45 ± 0.92 a
	1-Pentanol	0.3 ± 0.02 a	0.31 ± 0.01 a
	3-Methylpenta-1,3-diene-5-ol, (E)-	25.18 ±± 1.12 b	29.44 ± 1.08 a
	1-Hexanol	1.06 ± 0.51 b	10.05 ± 1.04 a
Aldehydes	Butanal, 3-methyl-	- b	0.09 ± 0.04 a
	Pentanal	0.35 ± 0.01 a	-b
	Hexanal	37.48 ± 0.88 a	36.49 ± 1.12 b
	2-Hexenal, (E)-	6.73 ± 0.86 b	9.66 ± 1.03 a
	2-hexenal	3.28 ± 0.05 a	0.45 ± 0.02 b
	(2E)-2-Ethyl-2-butenal	0.17 ± 0.01 a	- b
Ketones	1-Penten-3-one	0.08 ± 0.02 a	0.09 ± 0.02 a
	trans-3-Ethylidene-1-vinyl-2-pyrrolidone	1.23 ± 0.10 a	- b
	5-Hepten-2-one, 6-methyl-	0.06 ± 0.02 a	- b
Esters	Acetic acid, propyl ester	- b	0.13 ± 0.02 a
	Acetic acid, 2-methylpropyl ester	0.23 ± 0.01 a	0.42 ± 0.05 a
	Acetic acid, butyl ester	1.58 ± 0.51 a	1.29 ± 0.44 a
	1-Butanol, 2-methyl-, acetate	0.44 ± 0.01 a	- b
	1-Butanol, 3-methyl-, acetate	9.68 ± 1.04 a	- b
	Butanoic acid, propyl ester	0.08 ± 0.02 a	- b
	Propanoic acid, butyl ester	0.1 ± 0.02 a	0.07 ± 0.02 a
	Acetic acid, pentyl ester	0.07 ± 0.02 a	- b
	Butanoic acid, 2-methyl-, propyl ester	0.04 ± 0.01 a	- b
	2-Pentanol, propanoate	0.04 ± 0.01 a	- b
	Butanoic acid, butyl ester	0.07 ± 0.01 a	0.08 ± 0.01 a
	Acetic acid, hexyl ester	0.14 ± 0.01 a	0.15 ± 0.01 a
	Butyl 2-methylbutanoate	0.3 ± 0.02 a	0.15 ± 0.01 b
	Butanoic acid, 2-methyl-, hexyl ester	1.22 ± 0.19 a	0.39 ± 0.02 b
Hydrocarbons	7-Oxabicyclo [4.1.0]heptane	0.54 ± 0.10 a	- b
	Cyclohexene, 1-methyl-4-(1-methylethenyl)-, (S)-	0.04 ± 0.01 a	- b
	alpha.-Farnesene	0.13 ± 0.05 a	- b

Note: Different lowercase letters after the same row of data indicate that the difference between treatments was significant at the level of *p* ≤ 0.05; “-” means not detected.

## Data Availability

Not applicable.

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
