# Peer review of "Effects and Mechanism Analysis of Non-Bagging and Bagging Cultivation on the Growth and Content Change of Specific Substances of Fuji Apple Fruit"

_plants, 2023, doi:10.3390/plants12183309_

Round 1

Reviewer 1 Report

The authors investigate the effects and mechanism analysis of non-bagging and bagging cultivation on on the growth and content change of specific substances of Fuji apple fruit. Due to the importance of the non-bagging cultivation for the economy of the apple industry, the study is of great importance.

The results are well discussed. The differences and trends observed between the two treatments are highlighted and analyzed.

Other comments:

a) Line 173 - Please confirm if the figure number is accurate.

b) Line 288 - The SOD (superoxide dismutase), CAT (catalase), and POD (peroxidase) abbreviations are only defined on line 489. Please make the necessary corrections. Please check all the text

c) Line 386 - Is this phrase correct?

d) Line 456 - Please replace "physiological diseases" with "physiological disorders".

Author Response

Dear reviewer .

Please see the attachment,thank you.

Reviewer 2 Report

Line 16: first time use full farm “VC”

Revise last section of abstract, and add conclusion lines of the article

Major comments” very short introduction

Add more literature in introduction section and write down in a story farm

Write down aims of the study at the end of introduction

Line 88: P = 0.05 <

Line 111-120, combine into one paragraph

Line 125-131. combine into one paragraph

Line 136-140: combine into one paragraph

Line 141-150: combine into one paragraph

Line 155: first time use full farm “SPS”

Line 185-185: combine into one paragraph

Line 192-206: combine into one paragraph

Line 297: MDA. First time full farm explains

All enzymes name used full farm at first use

Discussion section is fine, but combine small paragraph into one paragraph. Revise carefully

Line 522-525: provide suitable reference

In conclusion section, revise it, add only key findings

English minor correction need,

Revise whole draft carefully.

minor revision needed

Author Response

Dear reviewer ,

Please see the attachment,thank you !

Round 2

Reviewer 2 Report

add conclusion lines, at the end of abstract

correct fig1 y axis, same as figure 2, 3 and 4 and check other figures

figure 10, quality very low

enzymes name define at first use, 

in the next, upload clean file without chnages, all changes must accept and new chnage done with red color,,, all track mode chnages accpt,

minor correction

Author Response

Dear reviewer,

Please see the attachment,thank you very much.
